# Hyperuricemia in Chronic Kidney Disease: Emerging Pathophysiology and a Novel Therapeutic Strategy

**DOI:** 10.3390/ijms26189000

**Published:** 2025-09-16

**Authors:** Tomoaki Takata, Yukari Mae, Shotaro Hoi, Takuji Iyama, Hajime Isomoto

**Affiliations:** Division of Gastroenterology and Nephrology, Faculty of Medicine, Tottori University, Yonago, Tottori 683-8504, Japan

**Keywords:** hyperuricemia, CKD, URAT1, SURI, renal underexcretion, glomerular under-filtration, tubular over-reabsorption, uric acid, uricosuric drug, urate excretion

## Abstract

Hyperuricemia has been increasingly recognized as a modifiable contributor to chronic kidney disease (CKD) progression. Although the traditional classification of hyperuricemia distinguished between renal underexcretion and renal overload types, recent studies suggest that hyperuricemia in patients with CKD can result from heterogeneous excretory defects, including glomerular under-filtration and tubular over-reabsorption. These distinct phenotypes may drive divergent renal injury mechanisms. Experimental and clinical data reveal that monosodium urate crystals and soluble uric acid independently induce renal damage through oxidative stress, inflammasome activation, and endothelial dysfunction. Furthermore, clinical investigations showed inconsistent associations between serum uric acid levels and renal outcomes, suggesting that serum levels alone may not fully reflect urate-related renal risk. This has prompted increasing interest in uricosuric agents, particularly the selective urate reabsorption inhibitors (SURIs), which target tubular urate handling. Urate transporter 1 inhibitors have shown promise in enhancing urinary uric acid excretion and potentially preserving kidney function, especially in patients with CKD. In this review, we summarize the current evidence linking the emerging pathophysiological classification of hyperuricemia, mechanisms or urate-induced kidney injury, and therapeutic interventions. These insights may inform individualized approaches to urate-lowering therapy in CKD and support future research into phenotype-guided treatment strategies.

## 1. Introduction

Chronic kidney disease (CKD), characterized by a progressive decline in renal function, is increasingly recognized as a global health issue. The prevalence of CKD continues to rise, particularly among aging populations and individuals with metabolic disorders [1,2]. CKD is associated not only with the risk of progression to end-stage kidney disease, but also with an elevated burden of cardiovascular disease [3,4]. The underlying causes of CKD are diverse, with diabetic nephropathy, nephrosclerosis, and chronic glomerulonephritis representing the most common etiologies worldwide. While each condition exhibits distinct histopathological and clinical features, they often converge on shared mechanisms, such as glomerulosclerosis, interstitial fibrosis, and vascular remodeling.

Recently, hyperuricemia has gained increasing attention as a potentially modifiable factor in CKD. In patients with impaired renal function, decreased uric acid excretion leads to elevated serum uric acid levels [5]. Beyond its well-established role in gout, asymptomatic hyperuricemia has been implicated in the pathogenesis of kidney damage through mechanisms involving oxidative stress, endothelial dysfunction, and intrarenal hemodynamic changes, ultimately leading to nephron loss [6]. Notably, recent insights suggest that hyperuricemia in CKD may reflect distinct excretory defects, broadly classified into impaired glomerular filtration and excessive tubular reabsorption [7,8]. These patterns may correspond to different vascular, interstitial, and glomerular pathologies and could help explain the heterogeneous clinical presentations seen in CKD patients with elevated serum uric acid levels.

In this review, we explore the evolving concept of hyperuricemia in CKD, focusing on its pathophysiological subtypes, mechanistic links to kidney damage, and the emerging therapeutic strategies tailored to these specific patterns of uric acid dysregulation. These perspectives may offer insight into the heterogeneous renal phenotypes observed in clinical practice.

## 2. Classification of Hyperuricemia in CKD

Uric acid is the terminal product of purine metabolism in humans, and its concentration in serum reflects a balance among production in the liver, intestinal excretion, and renal excretion [9]. The kidneys contribute most significantly to urate homeostasis, being responsible for the majority of daily uric acid disposal. Within the nephron, urate is freely filtered at the glomerulus and undergoes tubular reabsorption. The reabsorption is primarily mediated by urate transporter 1 (URAT1), expressed on the apical membrane of the proximal tubular epithelial cell, and glucose transporter 9 (GLUT9), expressed on the basolateral side. A smaller fraction of urate is secreted back into the urine through apical ATP-binding cassette transporter G2 (ABCG2) and basolateral organic anion transporters. This tightly regulated process results in the net excretion of about 10% of filtered urate under physiological conditions [7].

Traditionally, hyperuricemia has been classified into three categories based on its underlying pathophysiology: overproduction type, extrarenal underexcretion type, and renal underexcretion type. The first two are collectively referred to as renal overload types, while the latter is characterized by reduced renal urate clearance [9,10]. In patients with CKD, renal underexcretion is the most prevalent mechanism contributing to hyperuricemia, reflecting the impaired ability of the diseased kidney to eliminate uric acid efficiently.

However, recent investigation suggests that the renal underexcretion type is not homogenous. It may be more accurately described as comprising two distinct pathophysiological profiles. One subtype, termed glomerular under-filtration type, is characterized by a decreased glomerular filtration rate resulting in diminished urate delivery to the tubule. The other involves a relative excess tubular urate reabsorption in which urate handling is impaired independent of or disproportionate to GFR decline, and that can be termed tubular over-reabsorption type [7,8]. Differentiating between these two subtypes is important in terms of the pathophysiology of urate-mediated kidney injury. It addresses a central question whether elevated uric acid is merely a surrogate marker of renal dysfunction or whether it plays an active role in disease initiation. Patients with disproportionate tubular reabsorption may exhibit hyperuricemia even in early-stage CKD, and such individuals might represent a subset in which uric acid actively contributes to nephron injury. In contrast, when hyperuricemia arises solely in parallel with falling GFR, its role may be more reflective than causative.

Although no definitive diagnostic standard exists to separate these subtypes to date, certain biochemical indices may offer insight. Parameters such as the fractional excretion of uric acid (FEUA) and the urinary uric acid-to-creatinine ratio (UACR) may be candidates. A low FEUA may suggest excessive reabsorption, while an apparently normal FEUA in the setting of low GFR may mask a filtration deficit. Similarly, spot urine markers such as UACR could offer a practical alternative for assessing tubular urate handling. These indices remain to be established as clinical standards. Moreover, given the overlapping and dynamic nature of these excretory defects, the identification of more specific and sensitive indices beyond FEUA and UACR will be essential for accurately delineating pathophysiological subtypes. A composite tubulo-glomerular urate handling (TUH) index has been proposed to capture the relative contributions of filtration and tubular reabsorption to net urate excretion [7]: TUH = CCr × (100 − FEUA) = (CCr − CUA) × 100. Although this formulation remains theoretical and has not yet been validated, recognition of these pathophysiological subtypes underscores the complexity of urate handling in kidney disease and may provide a useful framework for future research as well as individualized therapeutic approaches. Prospective validation study of these indices and diagnostic criteria with defined performance characteristics are desired.

A conventional classification limited to “renal underexcretion” cannot distinguish whether hyperuricemia is a driver of kidney injury or a consequence of impaired excretion in CKD. Decomposing underexcretion into glomerular under-filtration and tubular over-reabsorption provides a testable framework to clarify directionality.

## 3. Mechanistic Links Between Hyperuricemia and CKD Progression

Hyperuricemia contributes to the development and progression of CKD through multiple mechanisms. These include direct injury to renal tubular cells, alterations in glomerular hemodynamics, and vascular endothelial dysfunction. While the deposition of monosodium urate (MSU) crystals is a well-recognized cause of kidney injury in gouty nephropathy, increasing evidence suggests that even in the absence of crystal formation, soluble uric acid can exert pathogenic effects on renal structures and function. These mechanisms may interact and vary depending on the underlying pattern of urate dysregulation.

### 3.1. Tubular Injury

Among the various renal structures affected by hyperuricemia, the renal tubules represent a primary site of injury. This can occur both through urate crystal-dependent and crystal-independent mechanisms.

MSU deposition can directly induce renal tubular injury [6]. While hyperuricemia is defined as serum uric acid exceeding 7.0 mg/dL, at which point MSU crystals may precipitate in the serum [11], crystallization in the kidney may occur independently, particularly within the tubular lumen. In this compartment, MSU crystals can form when the urinary environment becomes supersaturated with poorly soluble uric acid, especially under conditions of low pH and high local urate concentration. The risk of precipitation depends not on absolute excretion but on uric acid concentration relative to its solubility at a given pH. At a urine pH of 5.5, the solubility of undissociated uric acid is approximately 100 mg/L, with crystallization occurring when concentrations approach or exceed the metastable upper limit, estimated around 200 mg/L [12]. Once formed in the renal tubular lumen, MSU crystals may contribute to epithelial and interstitial injury via inflammatory mechanisms. A well-characterized pathway involves activation of the NOD-, LRR-, and pyrin domain–containing 3 (NLRP3) inflammasome [13]. This mechanism has been established primarily in the context of gouty arthritis, where MSU deposition in synovial tissue leads to uptake of crystals by monocytes and macrophages, triggering lysosomal rupture, potassium efflux, and caspase-1–dependent maturation of pro-inflammatory cytokines such as interleukin (IL)-1β and IL-18 [14]. Whether this cascade operates similarly in the kidney, particularly in renal tubular epithelial cells, remains unclear. However, MSU-dependent inflammasome activation has been proposed as a contributing mechanism in kidney injury models as well [15]. A recent in vivo study demonstrated that MSU crystals injected into the kidney directly induced inflammasome activation and interstitial fibrosis in the absence of hyperuricemia, supporting a potential pathogenic role of local MSU within the kidney [16]. Another in vivo investigation using genetically modified mice exhibiting hyperuricemia, crystalluria contributed to interstitial inflammation and fibrosis [17]. The route by which MSU is taken up into renal epithelial cells also remains to be clarified. In gouty arthritis, crystal uptake occurs via phagocytosis [18], but in the renal tubule, it is uncertain whether passive sedimentation, pinocytosis, or phagocytic-like mechanisms are involved. Taken together, while the paradigm of MSU-driven inflammasome activation is well established in gout, its translation to the kidney remains biologically plausible but mechanistically underexplored. Further studies using renal-specific models are needed to elucidate the precise role of MSU crystals in tubulointerstitial inflammation and progressive nephron injury.

In addition to crystal-induced mechanisms, soluble uric acid has been increasingly recognized as a potential mediator of renal tubular injury through crystal-independent pathways. Soluble uric acid has an effect on producing pro-inflammatory cytokine, including IL-1β in the absence of MSU crystallization [18,19,20]. NLRP3 inflammasome could be involved in soluble uric acid-induced kidney injury. Soluble uric acid induces inflammasome in proximal epithelial tubular cells in a TLR4-dependent manner [21]. A recent investigation highlighted that soluble uric acid can also activate the NLRP3 inflammasome through various stress-related mechanisms, including oxidative stress, mitochondrial dysfunction, reactive oxygen species (ROS) generation, lysosomal disruption, and endoplasmic reticulum stress, ultimately contributing to kidney injury [22]. Similarly, in cardiac myocytes, soluble uric acid has been shown to activate the NLRP3 inflammasome, leading to mitochondrial oxidative stress, caspase-1 activation, and subsequent cell injury [23]. These findings support the concept that both crystalline and non-crystalline forms of uric acid can exert deleterious effects on the renal tubules through distinct yet converging inflammatory and oxidative mechanisms. A better understanding of these mechanisms is essential for clarifying the bidirectional relationship between hyperuricemia and CKD (Figure 1).

### 3.2. Endothelial Dysfunction

All endothelial cells play a pivotal role in maintaining vascular homeostasis by regulating vasodilation, leukocyte trafficking, oxidative balance, and barrier integrity. Uric acid impairs endothelial function through multiple interrelated mechanisms. One major pathway involves the alteration in endothelium-dependent vasodilation due to the suppression of endothelial nitric oxide synthase (eNOS) activity, resulting in decreased nitric oxide (NO) bioavailability [24]. This is accompanied by increased production of reactive oxygen species (ROS), leading to oxidative stress and endothelial cell damage [25]. A study using human umbilical vein endothelial cells demonstrated that elevated extracellular uric acid concentrations upregulate GLUT9 expression, leading to increased intracellular urate accumulation and subsequent reduction in eNOS activity and NO production [26]. Consistent with these findings, serum nitrate levels, an indirect marker of NO production, were significantly decreased in a rat model of hyperuricemia. In addition, neuronal NO synthase expression was reduced in the renal medulla, suggesting that uric acid can suppress NO bioavailability in vivo as well [27]. These suppressed NO bioavailability and ROS generation lies mainstay in uric acid-induced endothelial dysfunction [28]. Furthermore, uric acid has been shown to stabilize and activate hypoxia-inducible factor-1α (HIF-1α), which induces vascular inflammation and increases endothelial permeability [29]. This promotes monocyte adhesion and transmigration, fostering a pro-inflammatory vascular environment. These processes promote structural remodeling of the arteriolar wall, including intimal thickening and diminished vascular compliance. In the kidney, endothelial dysfunction plays a key role in maintaining glomerular hemodynamics, by regulating vascular tone and autoregulatory responses at the afferent arteriole [30,31]. These changes may result in either glomerular hypoperfusion or glomerular hyperfiltration due to dysregulated vasodilation and impaired autoregulation. Both hemodynamic extremes can impose stress on the glomerular structure and contribute to kidney injury.

### 3.3. Glomerular Hemodynamic Alterations

Endothelial dysfunction is closely linked to abnormalities in glomerular hemodynamics. In the setting of hyperuricemia, uric acid–induced vascular injury affects the tone and structure of the afferent arteriole, leading to alterations in glomerular perfusion pressure. These changes in the afferent arteriole under hyperuricemia may manifest as two distinct clinical and pathophysiological patterns; glomerular hyperfiltration and hypoperfusion patterns.

Glomerular hyperfiltration pattern arises from afferent arteriolar dilation and impaired autoregulatory function. Hypertrophy and hyalinosis of the afferent arteriole can often be observed histologically. Clinical manifestation often presents with proteinuria as a marker of glomerular hyperfiltration. An intrarenal renin–angiotensin system (RAS) is also involved in this phenotype. Experimental models in rats demonstrated that hyperuricemia induces afferent arteriolar hyalinosis and glomerular hyperfiltration, in part through the activation of the intra intrarenal renin–angiotensin system (RAS), even in the absence of urate crystal deposition [32]. Consistent with these findings, in a human study, serum uric acid concentration was an independent predictor of blunted renal plasma flow response to angiotensin II, indicating an activated intrarenal RAS in subjects with high uric acid levels [33]. Persistent glomerular hyperfiltration imposes mechanical stress on podocytes and glomerular capillaries, resulting in urinary protein, glomerular structural change and progressive nephron loss [34,35].

Glomerular hypoperfusion pattern is characterized by narrowing of the afferent arteriole lumen and reduced blood flow to the glomerulus. This pattern is primarily considered to be driven by endothelial dysfunction, leading to vasoconstriction. As glomerular ischemia predominates in this setting, proteinuria is often absent or minimal. These two hemodynamic patterns reflect the pathological heterogeneity of CKD. Hyperfiltration-type injury is commonly associated with metabolic disorders such as diabetes mellitus and obesity, while hypoperfusion-type injury predominates in elderly individuals and those with arteriosclerosis or hypertensive nephrosclerosis. A seminal pathological study has described distinct afferent arteriolar lesion. Hyalinized medial hypertrophy with thickening and intimal multilayering with luminal collapse appear to correspond with these two phenotypes, reflecting underlying metabolic versus arteriosclerotic etiologies, respectively [36]. Furthermore, previous study conducted in patients who undergo renal biopsy demonstrated that both patterns of afferent arteriolar injury were significantly associated with elevated serum uric acid levels in patients with CKD [37]. These findings indicate that uric acid-associated glomerular hemodynamic alterations may underlie two distinct renal injury phenotypes with divergent clinical and histological features [38].

As most data reviewed here are preclinical or observational and therefore cannot, by themselves, prove causality in human CKD, these pathways should be interpreted as hypothesis-generating until clinical trials demonstrate renal benefit mediated through them (Figure 2).

## 4. Clinical Evidence of Hyperuricemia and CKD Progression

Numerous epidemiological studies have demonstrated a close relationship between elevated serum uric acid levels and both the onset and progression of CKD [39,40,41]. Even in the absence of gout, asymptomatic hyperuricemia has been associated with an increased risk of incident CKD and more rapid eGFR decline [42]. Several prospective cohort studies have consistently demonstrated that elevated serum uric acid levels independently predict an increased risk of CKD, eGFR decline, and the development of proteinuria [43,44]. A meta-analysis comprising over 577,000 individuals reported that each 1 mg/dL increment in baseline serum uric acid level was associated with a significant 15% higher risk of incident CKD [43]. Consistent observation has been reported in a five-year cohort of Japanese patients with type 2 diabetes, in which each 1 mg/dL increment of baseline serum uric acid was associated with a higher odds of rapid eGFR decline, even adjusted for multiple covariates [45]. These findings support a strong and independent association between higher serum uric acid and adverse renal outcomes. However, contradicting findings have also been reported, indicating that there is no relevant correlation between serum uric acid and renal outcomes [46,47].

These inconsistencies suggest that serum uric acid alone may not fully capture the complexity of urate-associated renal risk. Recent studies have turned attention to urinary uric acid excretion as an additional determinant of kidney prognosis. In a prospective analysis of CKD patients, lower urinary excretion of uric acid, assessed by UUCR and FEUA, were found to be independently linked to a higher risk of renal function decline [48]. Furthermore, in a cross-sectional study of CKD patients, urinary markers such as the FEUA and the UUCR, rather than serum uric acid levels or 24 h urinary excretion, were significantly associated with biomarkers of kidney damage [49] (Table 1).

The serum uric acid-to-creatinine ratio (SUCR) has also gained attention as a promising marker for urate burden adjusted for renal function. Serum uric acid levels are strongly influenced by renal function, making their interpretation challenging in patients with CKD. SUCR addresses this limitation by incorporating serum creatinine as a surrogate for GFR. A cross-sectional analysis of Japanese community-dwelling individuals demonstrated that higher SUCR values were significantly associated with the prevalence of hypertension, independent of serum uric acid levels [50]. In a large-scale prospective cohort, SUCR showed a nonlinear association with cardiovascular and all-cause mortality [51,52]. In a multicenter Italian cohort, an SUCR threshold of 5.35 was identified as a significant predictor of cardiovascular mortality in individuals with diabetes and preserved renal function [53]. Among participants with impaired renal function, an SUCR threshold of 7.50 was identified as independently associated with cardiovascular mortality [52]. Although these findings primarily address cardiovascular endpoints, they provide indirect support for SUCR as a potentially informative marker in renal outcome prediction. Given its ability to reflect uric acid burden relative to renal function, SUCR may serve as a useful biomarker for identifying CKD patients at heightened risk of progression. Further research is warranted to establish the optimal SUCR thresholds for predicting renal outcomes and to clarify its role in guiding individualized urate-lowering therapy in CKD populations.

## 5. Therapeutic Approaches to Hyperuricemia in CKD

Despite accumulating evidence linking hyperuricemia to the progression of CKD, the role of urate-lowering therapy (ULT) in improving renal outcomes remains an area of active investigation. Xanthine oxidase inhibitors (XOIs), such as alopurinol and febuxostat, have been most extensively studied. Some randomized controlled trials have reported renoprotective effects of XOIs, suggesting improvements in estimated GFR decline or proteinuria reduction [54,55]. However, other studies have failed to demonstrate consistent benefits. The febuxostat versus placebo randomized controlled trial regarding reduced renal function in Patients With Hyperuricemia Complicated by Chronic Kidney Disease Stage 3 (FEATHER) study found that febuxostat did not significantly slow eGFR decline compared to placebo [56]. Similarly, the preventing early real loss in diabetes (PERL) study, which evaluated allopurinol in patients with type 1 diabetes and diabetic kidney disease up to moderate stage, found no beneficial effect on the progression of kidney disease even with serum urate reduction [57]. Furthermore, the controlled trial of slowing of kidney disease progression from the inhibition of xanthine oxidase (CKD-FIX), a large randomized controlled trial enrolling patients with stage 3 or 4 CKD, demonstrated that allopurinol treatment failed to slow eGFR decline over two years compared to placebo, despite a sustained reduction in serum uric acid [58].

Although several large RCTs of XOIs were neutral for kidney outcomes, a recent study provided evidence supporting a potential renoprotective role of XOIs when a target serum urate level is achieved. In this study involving over 14,000 patients with gout and stage 3 CKD, of whom 98.8% were treated with allopurinol, patients who achieved a serum urate level below 6 mg/dL had a significantly lower risk of progression to advanced kidney disease compared to those who did not achieve this target [59]. A post hoc analysis of the FEATHER study showed that among CKD patients without proteinuria, those receiving febuxostat exhibited a significantly more favorable eGFR slope compared to those receiving placebo [60]. Furthermore, a post hoc analysis of the cardiovascular safety of febuxostat and allopurinol in patients with gout and cardiovascular morbidities (CARES) trial, which included over 5000 gout patients treated with febuxostat or allopurinol, showed that more than half of the patients maintained stable or improved eGFR throughout treatment. Notably, maintaining average serum urate levels below 6.0 mg/dL was independently associated with a reduced risk of eGFR decline [61]. These findings support the notion that individualized ULT, especially with attention to achieving urate targets, may play a protective role in renal function (Table 2).

## 6. Novel and Emerging Therapeutic Strategies

Uricosuric agents offer a mechanistically distinct approach from XOIs by increasing renal excretion of uric acid by inhibiting tubular reabsorption. This therapeutic strategy may offer a distinct advantage in CKD, where impaired renal excretion is a major contributor to hyperuricemia. A recent retrospective cohort study compared the renoprotective effects of three ULTs, including benzbromarone, allopurinol, and febuxostat, in CKD patients with hyperuricemia. Over a 13-year follow-up, benzbromarone use was associated with a significantly lower risk of progression to dialysis compared with allopurinol, despite febuxostat achieving greater serum urate reduction [62]. In contrast, a randomized controlled trial conducted in stage 3 CKD with hypertension and hyperuricemia, no significant differences could be observed in the effects of febuxostat and benzbromarone against renal function decline [63]. Furthermore, the use of benzbromarone in advanced CKD has been limited due to safety concerns, including the risk of hepatotoxicity and reduced efficacy in the setting of low GFR.

Recently, selective urate reabsorption inhibitor (SURI), which enhances the excretion of urinary uric acid by selectively inhibiting the URAT1 in the proximal tubule was developed as a new uricosuric agent [64,65,66,67,68,69,70,71,72,73,74,75,76,77,78]. Lesinurad, the earliest SURI developed, lowers serum uric acid level but clinical adoption was limited by renal adverse events and nephrolithiasis risk, and in the absence of established benefit on kidney outcomes, the agent was withdrawn. By contrast, dotinurad and verinurad have shown robust urate lowering effect in early phase RCTs. Across these trials, renal safety and kidney endpoints have generally been comparable to active or placebo comparators, but consistent renoprotective effects have not been established (Table 3). Therefore, these RCTs do not yet resolve whether URAT1 inhibition improves kidney outcomes.

To complement the limited trial evidence, we summarize real-world observations. Because dotinurad was approved earlier in Japan, most studies evaluating the renal effects of SURI therapy are retrospective, single-center investigations from Japanese cohorts (Table 4). Nevertheless, urate-lowering efficacy of dotinurad in patients with CKD has been consistently demonstrated across multiple studies [79,80]. Although its renoprotective effects remain variable, several studies have suggested potential renal benefits associated with URAT1 inhibition. In a retrospective observational study, dotinurad use was significantly associated with significant decrease in urinary albumin, and the increase in urinary uric acid excretion after dotinurad was negatively correlated with the change in serum creatinine, indicating that patients who exhibited greater increases in uric acid excretion tended to show less progression of renal dysfunction [81]. Similarly, in a retrospective study involving patients with varying degrees of CKD stages, dotinurad significantly reduced serum uric acid levels. A subgroup analysis revealed that patients with baseline eGFR < 30 mL/min/1.73 m^2^ exhibited a significant improvement in eGFR following dotinurad. Furthermore, multivariate logistic regression identified both eGFR < 30 mL/min/1.73 m^2^ and achieving serum uric acid levels ≤6.0 mg/dL as independent factors associated with eGFR improvement [82]. Consistently, dotinurad treatment led to a significant improvement in eGFR among CKD patients with hyperuricemia, with a positive correlation between the magnitude of serum urate reduction and the extent of eGFR improvement, supporting a potential renoprotective effect associated with enhanced urate excretion [83]. Moreover, when compared with febuxostat, only dotinurad demonstrated significant improvement in renal function despite achieving similar reductions in serum uric acid levels [84]. These findings indicate that enhancing urinary uric acid excretion may have renoprotective benefits that beyond serum uric acid reduction alone. It should be noted that clinicians should recognize the risks of nephrolithiasis and consider hydration and urine alkalinization in treating CKD patients with uricosuric agents.

## 7. Conclusions and Future Perspectives

Recent advances in our understanding of uric acid metabolism in CKD have highlighted the heterogeneous nature of hyperuricemia in this population. Hyperuricemia is now recognized to arise from distinct excretory patterns, glomerular under-filtration and tubular over-reabsorption. This conceptual shift underscores the need for a refined classification of hyperuricemia in CKD, not merely as a biochemical marker but as a dynamic contributor to disease pathogenesis. Identifying these subtypes may help clarify which patients are most vulnerable to urate-induced injury and which are most likely to benefit from targeted intervention.

The emergence of novel urate-lowering agents, particularly SURIs, offers promising avenues for precision medicine in hyperuricemic CKD. Unlike traditional XOIs, these agents act by enhancing urinary uric acid excretion, potentially addressing the pathophysiology of tubular reabsorption-dominant phenotypes. While further studies are required to establish causal relationships and treatment algorithms based on these subtypes, the accumulating clinical and mechanistic evidence supports a future in which urate-lowering therapy may be individualized, guided by specific biochemical profiles and renal excretory capacity. To operationalize these concepts, we provide a conceptual algorithm for phenotype-guided urate management (Figure 3).

As the objective of this review was to provide a focused and conceptual synthesis of the current evidence related to hyperuricemia subtypes, pathophysiological mechanisms, and emerging therapeutic approaches in CKD, this review was conducted as a narrative. While not exhaustive, this approach aimed to synthesize key findings and reflect current perspectives in this field. Because we focused on discussing a pathophysiology-based, phenotype-guided approach to urate handling in the kidney, we include SURI studies primarily as hypothesis-generating evidence. The absence of a consistent renal benefit of XOIs across several RCTs, despite substantial serum urate reductions, underscores unresolved issues: the limitations inherent to how urate is lowered, and whether serum urate alone is the optimal therapeutic target or surrogate for intrarenal urate burden. Mechanistic evidence for urate-mediated kidney injury, together with the observation that many patients with CKD exhibit an underexcretion phenotype supports evaluating strategies that directly address intrarenal urate handling. Prospective trials stratified by urate excretion phenotype will be essential to validate subtype-specific responses to therapy. By embracing a pathophysiology-based classification and leveraging novel therapeutics, clinicians may be better equipped to mitigate urate-mediated renal injury and improve outcomes for patients with CKD.

## Figures and Tables

**Figure 1 ijms-26-09000-f001:**
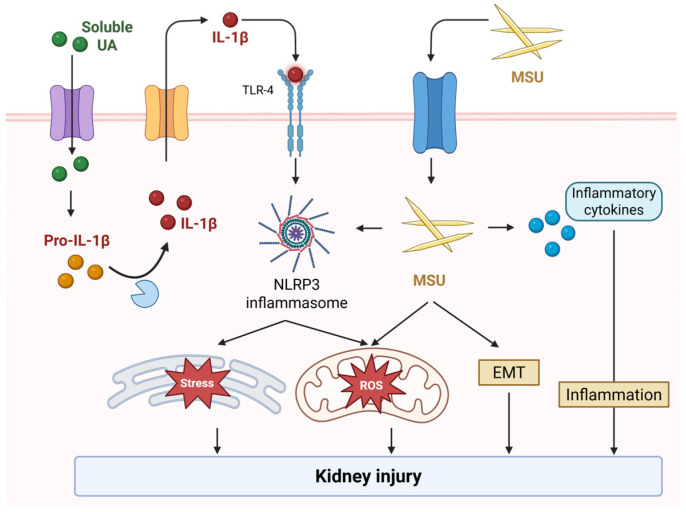
Mechanistic pathway of urate-mediated kidney injury. Soluble urate and MSU crystals both activate the NLRP3 inflammasome via TLR/NF-κB priming, cellular stress, or crystal uptake, leading to caspase-1-mediated IL-1β release. These signals converge on mitochondrial ROS, EMT, and tubular inflammation, culminating in kidney injury.

**Figure 2 ijms-26-09000-f002:**
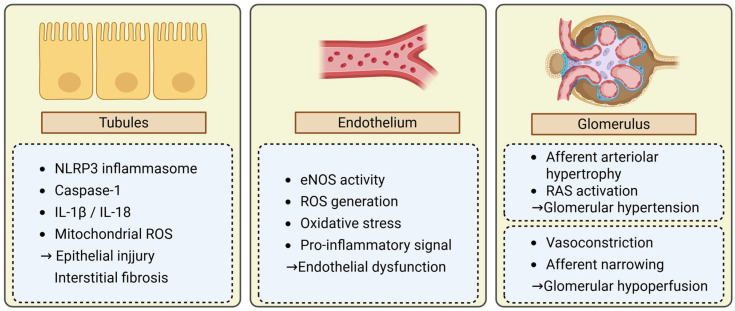
Mechanisms by which hyperuricemia may contribute to kidney injury. The schematic summary of three interconnected compartments. Tubules: urate related signals activate the NLRP3 inflammasome, leading to caspase-1 activation and IL-1β/IL-18 release. Mitochondrial ROS and stress responses amplify injury, culminating in epithelial cell injury and interstitial fibrosis. Epithelium: urate-mediated reduction in eNOS activity and NO bioavailability, increased ROS/oxidative stress, and pro-inflammatory signaling, resulting in endothelial dysfunction. Glomerulus: vascular changes include afferent arteriolar hypertrophy and intrarenal RAS activation, promoting glomerular hypertension; alternatively, vasoconstriction/afferent narrowing leads to glomerular hypoperfusion. eNOS, endothelial nitric oxide synthase; NO, nitric oxide; NLRP3, NOD-like receptor protein 3; RAS, renin–angiotensin system; ROS, reactive oxygen species.

**Figure 3 ijms-26-09000-f003:**
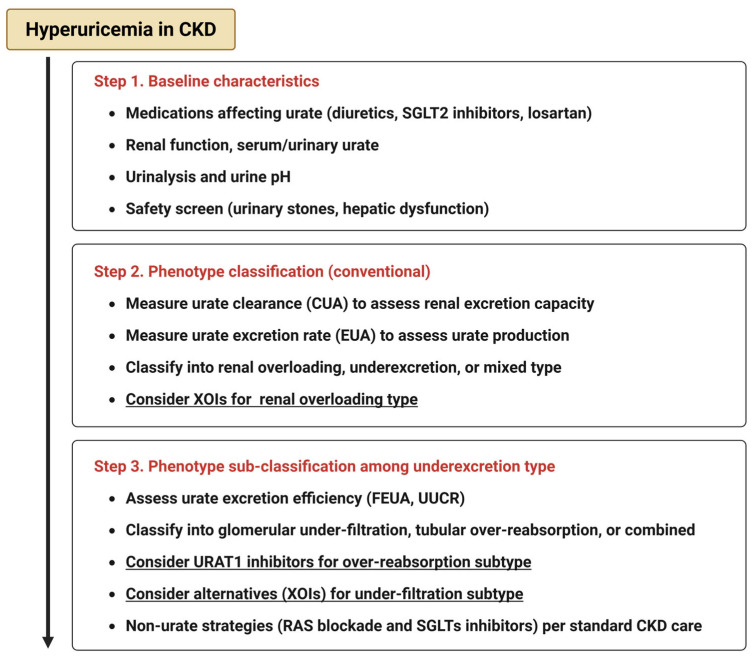
Conceptual algorithm for phenotype-guided urate management.

**Table 1 ijms-26-09000-t001:** Epidemiologic evidence linking hyperuricemia and CKD.

Ref.	Design	Population	Exposure	Kidney Outcome	Main Finding
[40]	Cross-sectional	T1D	SUA	eGFR decline	Higher SUA independently associated with lower eGFR
[42]	Retrospective	General	SUA	Incident CKD	Higher SUA is associated with new-onset CKD
[43]	Meta-analysis	General/CKD	SUA	Incident CKD	~15% higher CKD risk per 1 mg/dL higher SUA
[44]	Prospective	General/CKD	SUA	eGFR decline	~20% higher risk of eGFR decline per 1 mg/dL higher SUA
[45]	Prospective	T2D	SUA	Rapid eGFR decline	Higher SUA predicted faster decline in eGFR
[46]	Prospective	CKD G3-4	SUA	Renal failure	Neutral association
[47]	Prospective	CKD	SUA	Doubling of Cr/renal failure	Neutral association
[48]	Prospective	CKD	FEUA/UUCR	eGFR decline	Lower urinary urate excretion predicted eGFR decline
[49]	Cross-sectional	CKD	FEUA/UUCR	Kidney injury biomarkers	Urinary excretion of UA associated with injury markers

CKD, chronic kidney disease; T1D, type 1 diabetes; T2D, type 2 diabetes; SUA, serum uric acid; eGFR, estimated glomerular filtration rate; FEUA, fractional excretion of uric acid; UUCR, urinary uric acid-to-creatinine ratio.

**Table 2 ijms-26-09000-t002:** Therapeutic studies in CKD with xanthine oxidase inhibitors.

Ref.	Design	Population	Intervention	Kidney Outcome	Main Finding
[54]	RCT	CKD G3-4	Febuxostat	eGFR decline	Febuxostat slowed eGFR decline
[55]	RCT	CKD	Allopurinol	40% increase in Cr/dialysis	Allopurinol prevented composite renal endpoint
[56]	RCT	CKD G3	Febuxostat	eGFR slope	Neutral
[57]	RCT	T1D	Allopurinol	eGFR slope	Neutral
[58]	RCT	CKD G3-4	Allopurinol	eGFR slope	Neutral
[59]	Cohort	CKD G3	Mostly with allopurinol	Progression to advanced CKD	Target achievement associated with lower risk
[60]	Post hoc within RCT	CKD G3	Febuxostat	eGFR slope	Febuxostat slowed eGFR decline without proteinuria
[61]	Post hoc within RCT	Gout with CV risk	Febuxostat/allopurinol	eGFR trajectory	Lower SUA linked to less eGFR decline

RCT, randomized control trial; CKD, chronic kidney disease; eGFR, estimated glomerular filtration rate; SUA, serum uric acid.

**Table 3 ijms-26-09000-t003:** RCTs on URAT1 inhibitors.

Ref.	Intervention	Comparator	Duration	Renal Outcome/Safety
[64]	Lesinurad + allopurinol	Placebo + allopurinol	4 weeks	Renal parameters neutral
[65]	Lesinurad + allopurinol	Placebo + allopurinol	12 months	Renal AEs and sCr ≥ 1.5× more frequent with 400 mg
[66]	Lesinurad + allopurinol	Placebo + allopurinol	12 months	Renal AEs and sCr ≥ 1.5× higher with 400 mg
[67]	Lesinurad + febuxostat	Febuxostat + placebo	12 months	Renal AEs increased with 400 mg
[68]	Lesinurad	Placebo	6 months	Renal AEs more frequent with 400 mg
[69]	Verinurad + febuxostat	Febuxostat, verinurad, benzbromarone	42 days	No concerning renal signal reported
[70]	Lesinurad + febuxostat	No placebo	12 months	No new renal safety signals
[71]	Verinurad	Placebo	24 weeks	sCr elevations more frequent with verinurad
[72]	Dotinurad	Fabuxostat	14 weeks	AE rates comparable
[73]	Dotinurad	Placebo	8 weeks	Overall renal function stable
[74]	Dotinurad	Placebo	12 weeks	Consistent renal safety
[75]	Dotinurad	Benzbromarone	14 weeks	AE rates comparable
[76]	Verinurad + febuxostat	Placebo	24 weeks	Verinurad reduced UACR, overall safety similar
[77]	Verinurad + allopurinol	Alopurinol/placebo	34 weeks	No improvement in UACR or eGFR decline
[78]	Verinurad + allopurinol	Alopurinol/placebo	32 weeks	No between-group differences in eGFR and UACR

**Table 4 ijms-26-09000-t004:** Therapeutic studies in CKD with SURIs.

Ref.	Design	Population	Intervention	Kidney Outcome	Main Finding
[79]	Prospective	CKD G1-4	Dotinurad	eGFR change	eGFR tended to improve
[80]	Retrospective	CKD G3-5	Dotinurad	eGFR slope	eGFR slope improved
[81]	Retrospective	CKD	Dotinurad	Albuminuria, Cr	Increase in urate excretion correlated with less Cr rise
[82]	Retrospective	CKD	Dotinurad	eGFR change	eGFR improved in eGFR < 30 subgroup
[83]	Retrospective	CKD	Dotinurad	eGFR change	eGFR improved
[84]	Retrospective	CKD	Dotinurad vs. febuxostat	eGFR change	Only dotinurad group showed eGFR improvement

CKD, chronic kidney disease; eGFR, estimated glomerular filtration rate.

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
