# Peer review of "Hyperuricemia in Chronic Kidney Disease: Emerging Pathophysiology and a Novel Therapeutic Strategy"

_ijms, 2025, doi:10.3390/ijms26189000_

Round 1

Reviewer 1 Report

Comments and Suggestions for Authors

Specific recommendations for the revision:

  • In the Introduction (line no. 51): Provide the full form of “SUA”.
  • In the Therapeutic approaches: Authors have mentioned that “Some randomized controlled trials have reported renoprotective effects of XOIs, suggesting improvements in estimated GFR decline or proteinuria reduction. However, other studies have failed to demonstrate consistent benefits.” Provide appropriate references for these statements.
  • It is highly recommended to provide a detailed table listing various clinical studies that indicate “Hyperuricemia in CKDs” as well as the therapeutic-clinical studies of urate reabsorption inhibitors (SURIs) and Xanthine oxidase inhibitors (XOIs) in such patients.
  • Also, incorporate a detailed figure illustrating the molecular mechanism underlying “Hyperuricemia in CKDs”.

Author Response

Specific recommendations for the revision:

In the Introduction (line no. 51): Provide the full form of “SUA”.

Response: we have revised SUA to serum uric acid.

In the Therapeutic approaches: Authors have mentioned that “Some randomized controlled trials have reported renoprotective effects of XOIs, suggesting improvements in estimated GFR decline or proteinuria reduction. However, other studies have failed to demonstrate consistent benefits.” Provide appropriate references for these statements.

Response: Thank you for this comment. Appropriate references were added.

It is highly recommended to provide a detailed table listing various clinical studies that indicate “Hyperuricemia in CKDs” as well as the therapeutic-clinical studies of urate reabsorption inhibitors (SURIs) and Xanthine oxidase inhibitors (XOIs) in such patients.

Response: Thank you very much for this constructive advice. We have added Tables describing (1)epidemiologic evidence linking hyperuricemia and CKD, (2)therapeutic studies with XOIs, and (3)therapeutic studies with SURIs.

Also, incorporate a detailed figure illustrating the molecular mechanism underlying “Hyperuricemia in CKDs”.

Response: Thank you for this advice. We have added a figure illustrating the molecular mechanism underlying hyperuricemia in CKD.

Reviewer 2 Report

Comments and Suggestions for Authors

How does this review advance the field beyond your own prior publications on the same classification scheme ([7,8])? Can you specify what is genuinely novel in this paper?
Given that SURIs and dotinurad have been discussed in earlier literature, including by your group, why is this presented as a “novel therapeutic strategy” in the title?
What was your literature search methodology? Without this, how can readers be assured that the included studies were not selectively chosen to support your hypothesis?
Why does the review rely so heavily on Japanese cohorts and studies, often retrospective and single-center, without a parallel appraisal of global evidence?
How do you account for the multiple large RCTs (e.g., PERL, CKD-FIX, FEATHER) showing no benefit of XOIs, and why are these results downplayed compared to smaller observational studies showing benefit?
You frequently present mechanistic pathways (e.g., NLRP3 activation, endothelial NO suppression) as if causality in human CKD is established. What is your basis for asserting causality given the lack of consistent clinical trial confirmation?
How do you address the possibility of reverse causation (CKD leading to hyperuricemia rather than the reverse) in interpreting the epidemiological data?
What objective, validated criteria exist for distinguishing “glomerular under-filtration” from “tubular over-reabsorption” phenotypes in clinical practice?
FEUA and UACR are proposed as diagnostic aids, yet you acknowledge no standards exist. How can clinicians reliably use these measures without sensitivity/specificity data?
Have you tested whether phenotype-guided treatment actually improves outcomes compared to standard urate-lowering therapy in unstratified CKD populations?
The review disproportionately emphasizes dotinurad compared to other agents. How can you guarantee that this does not represent promotional bias, as you also had previous knowledge of this medicine? 
What are the possible safety issues (e.g., uricosuric-induced nephrolithiasis) of promoting uricosuric therapy in the CKD cirrhosis population, and why are they not stated in detail?
Why is no practical treatment algorithm provided if the goal is to advocate for phenotype-guided therapy?
Given your close research involvement with dotinurad, how do you address potential subconscious bias in the framing of the therapeutic section?

Author Response

How does this review advance the field beyond your own prior publications on the same classification scheme ([7,8])? Can you specify what is genuinely novel in this paper?

Response: Thank you for this comment. While our earlier works introduced the conceptual classification of hyperuricemia in CKD into glomerular under-filtration and tubular over-reabsorption types, this review substantially extends and deepens that framework. In this review, we provide a comprehensive synthesis of recent experimental and clinical findings that mechanistically link each hyperuricemia subtype to distinct pathways of kidney injury. In particular, we integrate emerging evidence related to endothelial dysfunction and glomerular hemodynamic alterations that were not fully addressed in our prior work. Furthermore, this paper proposes and critically evaluates biochemical indices such as FEUA, UUCR as tools to infer underlying urate handling defects. This paper also introduced SUCR as key index to assess urate dynamics in CKD patients. Although, in part, building on our previous conceptual proposal, this review offers significant advances by expanding its clinical relevance.

Given that SURIs and dotinurad have been discussed in earlier literature, including by your group, why is this presented as a “novel therapeutic strategy” in the title?

Response: Thank you for this comment. We agree that SURIs and dotinurad have been previously described. However, in the context of this review, we use the term “novel” to emphasize their emerging relevance as a strategic shift in the therapeutic approach to hyperuricemia in CKD, particularly in light of the recently proposed classification into tubular over-reabsorption and glomerular under-filtration types.

What was your literature search methodology? Without this, how can readers be assured that the included studies were not selectively chosen to support your hypothesis?

Response: Thank you very much for this comment. As this manuscript was structured as a narrative review, our objective was to provide a focused and conceptual synthesis of the current evidence related to hyperuricemia subtypes, pathophysiological mechanisms, and emerging therapeutic approaches in CKD. We did not perform a formal systematic search with predefined inclusion and exclusion criteria. Instead, the cited literature was selected based on its relevance to the central theme, including seminal experimental findings, representative clinical trials, and recent advances in urate-lowering therapies. We have added a brief clarification regarding the narrative nature of the review and the rationale for literature selection in the revised manuscript as follows; “As the objective of this review was to provide a focused and conceptual synthesis of the current evidence related to hyperuricemia subtypes, pathophysiological mechanisms, and emerging therapeutic approaches in CKD, this review was conducted as a narrative. While not exhaustive, this approach aimed to synthesize key findings and reflect current perspectives in this field.”

Why does the review rely so heavily on Japanese cohorts and studies, often retrospective and single-center, without a parallel appraisal of global evidence?

Response: Thank you for this comment. Our emphasis on SURI reflects where this drug class is currently in clinical use. Dotinurad has been approved and marketed in Japan preceding to the other countries. Furthermore, reports investigating the effect of the other SURIs such as lesinurad and verinurad on renal function is limited. Consequently, early post-marketing datasets presently originate from Japanese centers. While we agree that the evidence for SURIs differs in design from that for XOIs. However, inconsistent results of XOIs for renoprotective effects, recent advances clarifying crystal-dependent and crystal-independent urate injury pathways, and an emerging classification of hyperuricemia, provide a rationale for therapy tailored to intrarenal urate handling. As outlined in the preceding sections, this phenotype-guided approach represents a novel conceptual framework rather than a simple extension of prior XOI data. We therefore present SURI findings as hypothesis-generating, explicitly label their geographic and retrospective single-center nature in our manuscript. We have added following statement “Because dotinurad was approved earlier in Japan, most studies evaluating the renal effects of SURI therapy are retrospective, single-center investigations from Japanese cohorts.”

How do you account for the multiple large RCTs (e.g., PERL, CKD-FIX, FEATHER) showing no benefit of XOIs, and why are these results downplayed compared to smaller observational studies showing benefit?

Response: Thank you for this comment. The scope of our review is to discuss a pathophysiology-based, phenotype-guided approach to urate handling in the kidney. XOIs target production, whereas selective SURIs target tubular reabsorption and thereby intrarenal urate handling. Although our intent is not to downplay the multiple large RCTs of XOIs, the absence of a consistent renal benefit across these RCTs, despite substantial serum urate reductions, underscores several unresolved issues: the limitations inherent to how urate is lowered (production blockade vs. acceleration of urinary excretion), and whether serum urate alone is the optimal therapeutic target or surrogate for intrarenal urate burden. Mechanistic evidence for urate-mediated kidney injury, together with the observation that many patients with CKD exhibit an underexcretion phenotype supports evaluating strategies that directly address intrarenal urate handling. While we fully agree that the current SURI literature is less mature and largely hypothesis-generating, our argument is that a pathophysiology-based classification of hyperuricemia can enable phenotype-guided therapy, including SURI use, as a novel treatment strategy. This raises the need for multicenter, multi-region, and prospective trial to establish generalizability of our hypothesis. We have incorporated these into the conclusion and future perspectives.

You frequently present mechanistic pathways (e.g., NLRP3 activation, endothelial NO suppression) as if causality in human CKD is established. What is your basis for asserting causality given the lack of consistent clinical trial confirmation?

Response: Thank you for this point. We agree that causality in human CKD has not been established. Our mechanistic sections were intended to summarize biologic plausibility derived from experimental and translational studies rather than to assert proven causal pathways in humans. To avoid implication of established causality, we have added the statement as follows; “As most data reviewed here are preclinical or observational and therefore cannot, by themselves, prove causality in human CKD, these pathways should be interpreted as hypothesis-generating until clinical trials demonstrate renal benefit mediated through them.”

How do you address the possibility of reverse causation (CKD leading to hyperuricemia rather than the reverse) in interpreting the epidemiological data?

Response: This is an important point. A conventional classification limited to “renal underexcretion” cannot distinguish whether hyperuricemia is a driver of kidney injury or a consequence of CKD. While observational studies alone are unlikely to resolve this, more detailed phenotyping that separates glomerular under-filtration from tubular over-reabsorption offers a testable framework to clarify directionality and, ultimately, to enable phenotype-enriched trials that address causality. We have incorporated these into the “Classification of hyperuricemia in CKD” section.

What objective, validated criteria exist for distinguishing “glomerular under-filtration” from “tubular over-reabsorption” phenotypes in clinical practice?

Response: Thank you for this comment. At present, there are no validated, consensus clinical criteria that objectively distinguish “glomerular under-filtration” from “tubular over-reabsorption.” As noted in Section 2, FEUA and UUCR can serve as exploratory markers. In addition, a composite tubulo-glomerular urate handling (TUH) index has been proposed from our group (PMID: 40707231) to integrate filtration and tubular reabsorption:

TUH = CCr × (100 − FEUA) = (CCr − CUA) × 100, with FEUA = 100 × (CUA/CCr) (CCr = creatinine clearance; CUA = urate clearance). This formulation is theoretical and requires validation, but it provides a framework to operationalize phenotype separation in future studies.

FEUA and UACR are proposed as diagnostic aids, yet you acknowledge no standards exist. How can clinicians reliably use these measures without sensitivity/specificity data?

Response: Thank you for this comment. We agree that, in the absence of validated thresholds and sensitivity/specificity data, FEUA and UUCR cannot be used as diagnostic tests to classify “glomerular under-filtration” versus “tubular over-reabsorption.” Although this limitation is noted in our original manuscript, we clarified that these measures need prospective validation with defined performance characteristics in Section 2.

Have you tested whether phenotype-guided treatment actually improves outcomes compared to standard urate-lowering therapy in unstratified CKD populations?

Response: Thank you for this important comment. Phenotype-guided treatment has not yet been prospectively evaluated versus standard, unstratified therapy and should be considered a future direction in a “research article”.

The review disproportionately emphasizes dotinurad compared to other agents. How can you guarantee that this does not represent promotional bias, as you also had previous knowledge of this medicine?

Response: Thank you very much for this comment. We understand your concern and carefully revised the manuscript to minimize promotional appearance. Our emphasis on dotinurad reflects the current evidence landscape; it is presently the major marketed SURI and most clinical reports come from Japan, as we noted in our prior response. In the revised version, we reframed the text at the drug class level, using SURIs/URAT1 inhibition as the organizing concept. We also acknowledge that we have no conflict of interest regarding this manuscript.

What are the possible safety issues (e.g., uricosuric-induced nephrolithiasis) of promoting uricosuric therapy in the CKD cirrhosis population, and why are they not stated in detail?

Why is no practical treatment algorithm provided if the goal is to advocate for phenotype-guided therapy?

Response: Thank you for this comment. This review does not advocate routine uricosuric (or dotinurad) therapy, but our purpose is to discuss the possibility of a phenotype-guided framework for intrarenal urate handling. In this context, we note that SURI could be relevant to a putative tubular over-reabsorption type. Because we are not promoting uricosuric therapy, detailed drug-specific safety guidance was outside our scope. Nevertheless, to avoid ambiguity we have added a brief safety issue in CKD. Hydration and urine alkalinization need to be considered to avoid nephrolithiasis. We added safety note as follows; “It should be noted that clinicians should recognize the risks of nephrolithiasis and con-sider hydration and urine alkalinization in treating CKD patients with uricosuric agents.” We do not provide a practical treatment algorithm. As we state in Section 2 and the Conclusion, validated criteria to differentiate glomerular under-filtration from tubular over-reabsorption do not exist. A prescriptive algorithm at this time would risk premature guidance. We therefore propose phenotype-guided therapy as a research agenda, calling for future trials.

Given your close research involvement with dotinurad, how do you address potential subconscious bias in the framing of the therapeutic section?

Response: Thank you for this comment. We agree that the involvement of prior researches with dotinurad can introduce potential subconscious bias. As we noted in our prior response, we revised the manuscript to minimize the risk;

We revised the therapeutic section using URAT1 inhibition as a drug class, with dotinurad cited as the most reported example. We present limitations, particular for safety concerns, alongside positive results and we emphasize that evidence supporting our mechanism-based subclassification remains to be validated.

Round 2

Reviewer 1 Report

Comments and Suggestions for Authors

Authors have satisfactorily revised the manuscript. 

However, in "Table 2. Therapeutic studies in CKD with xanthine oxidase inhibitors" for references 59 and 61 authors didn't mention the name of drug.

In reference 61, it was noted that "ULT with either febuxostat or allopurinol for more than six months", hence revise the intervention.

For reference 59, mention the name of the drug; otherwise, it is recommended to remove the reference.

Author Response

Authors have satisfactorily revised the manuscript.

However, in "Table 2. Therapeutic studies in CKD with xanthine oxidase inhibitors" for references 59 and 61 authors didn't mention the name of drug.

Response: Thank you for this comment. In reference 59 study, 98.8% of the patients were treated with allopurinol. In reference 61 study, 50% were treated with febuxostat and the remaining 50% were with allopurinol. Therefore, we included these studies into Table 2.

In reference 61, it was noted that "ULT with either febuxostat or allopurinol for more than six months", hence revise the intervention.

Response: Thank you for this comment. We have revised the intervention.

For reference 59, mention the name of the drug; otherwise, it is recommended to remove the reference.

Response: Thank you for this comment. We have added the name of the drug.

Reviewer 2 Report

Comments and Suggestions for Authors

You state that this review "extends and deepens" the classification scheme presented in your prior work ([7,8]), but much of the conceptual framing appears unchanged. Could you specify precisely which mechanistic insights, clinical tools, or therapeutic implications are presented in this review?

You propose using FEUA, UUCR, and SUCR as potential markers for urate handling subtypes, yet acknowledge that they are not validated. Can you outline a plan or cite any preliminary data that would support the future validation of these markers in relation to clinical outcomes or treatment response?

Given your aim of promoting phenotype-guided therapy, could you provide at least a conceptual treatment framework or visual algorithm to illustrate how this might work in clinical practice, even if only as a hypothesis-generating proposal?

What is your plan to manage the generalizability of your proposed classification and treatment strategy, given that most of the supporting evidence comes from retrospective Japanese cohorts? Have you considered external validation using other datasets or registries?

Despite your efforts to reframe dotinurad as an example within the URAT1 inhibitor class, the manuscript still appears to disproportionately emphasize this drug. Can you revise the therapeutic section to provide a more balanced class-level analysis, including both promising and negative data for other uricosurics?

You argue that the failure of XOI trials may be due to mismatched treatment targets. However, could you provide a more detailed and structured analysis comparing these large RCTs (e.g., PERL, CKD-FIX) to your proposed phenotype-based approach? How would a phenotype-stratified trial design address the limitations seen in these RCTs?

Even though this is a narrative review, could you elaborate on your literature search strategy? For example, what databases were searched, what key terms and synonyms, if any, you used, and what years were included? This information would provide readers with a sense of the objectivity and breadth of your evidence synthesis.

Author Response

You state that this review "extends and deepens" the classification scheme presented in your prior work ([7,8]), but much of the conceptual framing appears unchanged. Could you specify precisely which mechanistic insights, clinical tools, or therapeutic implications are presented in this review?

Response: Thank you for this comment. We have clarified how the present review extends and deepens our prior classification work. For mechanistic insights, we separate crystal-dependent and crystal-independent tubular pathways. Furthermore, we summarized the plausible mediators, such as NLRP3 inflammasome, interleukins, and ROS, involved in the urate-induced kidney injury. We also integrate endothelial biology, such as reduced eNOS/NO, oxidative stress, and HIF, with two afferent-arteriolar patterns to explain heterogeneous renal injury. A new figure has been added to visualize these links between urate and renal injury mechanisms. For clinical tools, beyond FEUA and UUCR we introduce the tubulo-glomerular urate handling (TUH) index as a composite descriptor of filtration vs tubular reabsorption, and we summarize SUCR as a renal urate-burden marker normalized to kidney function. We emphasize that these indices are not yet validated for phenotype assignment; we present them as hypothesis-generating and outline their use as targets for future validation. Regarding therapeutic implications, prior work proposed a subclassification but did not connect subtypes to management. Here we take the next step by raising the hypothesis that patients within the broad “renal underexcretion” category comprise sub-phenotypes with different renal risks and potentially different treatment responses, such that URAT1 inhibition may be particularly relevant to a tubular over-reabsorption phenotype. While further investigation is still required, we argue that treatment selection should be considered in relation to subtype and we provide a non-prescriptive workflow and safety notes to guide hypothesis-driven study design rather than current practice. Importantly, incorporating the over-reabsorption concept shifts the focus beyond serum urate alone to include urinary urate as a practical marker. This reframes the field from a “hyperuricemia–CKD” to an “urate excretion impairment–CKD” framework, with potential to refresh treatment eligibility and trial design

You propose using FEUA, UUCR, and SUCR as potential markers for urate handling subtypes, yet acknowledge that they are not validated. Can you outline a plan or cite any preliminary data that would support the future validation of these markers in relation to clinical outcomes or treatment response?

Response: Thank you for this suggestion. We agree that not all indices are at the same stage of development. In our proposal, FEUA (equivalently TRUA) is a physiology-oriented measure of net tubular urate handling, analogous to the use of FENa for tubular sodium handling. Accordingly, we use FEUA as the operational reference (gold standard) for the concept of tubular over-reabsorption and CCr as the gold standard for the glomerular filtration rate, recognizing that what requires further work is not validating FEUA’s physiologic meaning but rather reaching consensus on decision thresholds if they are to be used clinically. By contrast, UUCR and SUCR require formal validation as surrogate markers. Our plan is: 1) define FEUA-based phenotypes as the reference and evaluate concordance of UUCR/SUCR with this phenotype (AUC, calibration) under standardized sampling and with sensitivity analyses. 2) Assess whether UUCR/SUCR add incremental prediction for eGFR slope and kidney composites. As preliminary data, we have included a previous investigation that showed low UUCR as independent determinant for a higher risk of renal function decline [48].

Given your aim of promoting phenotype-guided therapy, could you provide at least a conceptual treatment framework or visual algorithm to illustrate how this might work in clinical practice, even if only as a hypothesis-generating proposal?

Response: Thank you for this helpful suggestion. We agree that a conceptual, hypothesis-generating framework will aid readers. We have added a visual algorithm that illustrates how phenotype-guided therapy could be applied (Figure 3).

What is your plan to manage the generalizability of your proposed classification and treatment strategy, given that most of the supporting evidence comes from retrospective Japanese cohorts? Have you considered external validation using other datasets or registries?

Response: Thank you for highlighting this limitation. We agree that most supportive evidence to date comes from retrospective Japanese cohorts. To address generalizability, we outline a two-pronged plan. First, external validation across regions and care settings will be required. This includes investigations of our framework in non-Japanese datasets/registries (North America, Europe, and Asia-Pacific) that include diverse CKD etiologies (diabetic, hypertensive, glomerular) and ancestries. Second, evaluations in longitudinal cohorts whether mechanism-based subclassification and supportive indices predict eGFR slope and a kidney composite beyond eGFR.

Despite your efforts to reframe dotinurad as an example within the URAT1 inhibitor class, the manuscript still appears to disproportionately emphasize this drug. Can you revise the therapeutic section to provide a more balanced class-level analysis, including both promising and negative data for other uricosurics?

Response: Thank you very much for this comment. We wish to clarify that our review does not take a position against XOIs. Rather, based on the most recent evidence, we recognize that renoprotection with XOIs may be achievable in selected contexts, particularly when serum urate targets are attained and patient phenotype is favorable. In Section 5 “Therapeutic approaches to hyperuricemia in CKD” we deliberately present a balanced synthesis of the XOI evidence. The first paragraph summarizes the large RCTs showing that allopurinol or febuxostat did not consistently slow eGFR decline despite effective serum lowering. However, in the second paragraph, we highlighted more recent analysis that suggest benefit when treatment actually achieves a urate target or when clinical context is favorable; (i) in >14,000 patients with gout and stage 3 CKD (predominantly on allopurinol), maintaining serum urate <6 mg/dL associated with lower risk of progression [59]; (ii) in a post-hoc FEATHER analysis, non-proteinuric patients on febuxostat had a more favorable eGFR slope [60]; and (iii) in post-hoc CARES, patients who maintained serum urate <6 mg/dL more often had stable eGFR [61]. These observations support our view that XOIs can remain a viable therapeutic option in CKD. We have changed to “Although several large RCTs of XOIs were neutral for kidney outcomes, a recent study provided evidence supporting a potential renoprotective role of XOIs…” in Section 5 “Therapeutic approaches to hyperuricemia in CKD”.

In addition to the new Table 3 summarizing RCTs on URAT1 inhibitors, we revised the section 6 “Novel and emerging therapeutic strategies” The following description “Given the inconsistent renal benefits observed with conventional XOIs, attention has shifted toward uricosuric agents…” has removed and the other SURIs including lesinurad and verinurad were also introduced as follows “Recently, selective urate reabsorption inhibitor (SURI), which enhances the excretion of urinary uric acid by selectively inhibiting the URAT1 in the proximal tubule was developed as a new uricosuric agent [64-78]. Lesinurad, the earliest SURI developed, lowers serum uric acid level but clinical adoption was limited by renal adverse events and nephrolithiasis risk, and in the absence of established benefit on kidney outcomes, the agent was withdrawn. By contrast, dotinurad and verinurad have shown robust urate lowering effect in early phase RCTs. Across these trials, renal safe-ty and kidney endpoints have generally been comparable to active or placebo comparators, but consistent renoprotective effects have not been established (Table 3). There-fore, these RCTs do not yet resolve whether URAT1 inhibition improves kidney out-comes.”

You argue that the failure of XOI trials may be due to mismatched treatment targets. However, could you provide a more detailed and structured analysis comparing these large RCTs (e.g., PERL, CKD-FIX) to your proposed phenotype-based approach? How would a phenotype-stratified trial design address the limitations seen in these RCTs?

Response: Thank you for this comment. To avoid over-interpretation, we have removed the sentence “These findings suggest that the overall benefit of XOIs in slowing CKD progression may be limited in unselected populations”, because it could be misleading. A phenotype-based analysis of prior XOI RCTs might be intrinsically constrained, as tubular urate-handling metrics were not uniformly collected in those trials. However, we note that definitive testing requires prospective collection. Beyond this wording change, we made the following revisions to address your advice; (i) we added a conceptual algorithm (Figure 3) illustrating how phenotype-guided treatment could be applied, (ii) we revised the therapeutic section to present a class-level, neutral synthesis of URAT1 inhibition (including lesinurad, dotinurad, and verinurad), summarizing positive, neutral, and negative findings, (iii) we added concise evidence of SURI trials to facilitate comparison across designs and kidney outcomes (Table 3).

Even though this is a narrative review, could you elaborate on your literature search strategy? For example, what databases were searched, what key terms and synonyms, if any, you used, and what years were included? This information would provide readers with a sense of the objectivity and breadth of your evidence synthesis.

Response: We added a brief search strategy. For SURIs, we ran a structured PubMed search limited to Clinical Trial records (humans/adults; English; 2000–Aug 2025) using “lesinurad OR verinurad OR dotinurad”, extracting baseline renal function and kidney outcomes/safety. For XOIs, we used a focused, non-structured PubMed search with “allopurinol OR febuxostat” plus CKD/kidney-outcome terms to summarize the major RCTs and key follow-ups. We note this asymmetric approach and note it reflects the review’s emphasis on the tubular (URAT1) pathway while aiming for balance.

Round 3

Reviewer 2 Report

Comments and Suggestions for Authors

The paper can be accepted in its present form.